# AccVideo: Accelerating Video Diffusion Model with Synthetic Dataset

## Abstract

Diffusion models have achieved remarkable progress in the field of video generation. However, their iterative denoising nature requires a large number of inference steps to generate a video, which is slow and computationally expensive. In this paper, we begin with a detailed analysis of the challenges present in existing diffusion distillation methods and propose a novel efficient method, namely **AccVideo**, to reduce the inference steps for accelerating video diffusion models with synthetic dataset. We leverage the pretrained video diffusion model to generate multiple valid denoising trajectories as our synthetic dataset, which eliminates the use of useless data points during distillation. Based on the synthetic dataset, we design a trajectory-based few-step guidance that utilizes key data points from the denoising trajectories to efficiently learn the noise-to-video mapping, enabling video generation in fewer steps. Furthermore, since the synthetic dataset captures the data distribution at each intermediate diffusion timestep, we introduce an adversarial training strategy to align the intermediate distribution of the student model with that of our synthetic dataset, thereby enhancing the video quality. Extensive experiments demonstrate that our model achieves 8.5× improvement in generation speed compared to the teacher model while maintaining comparable performance. Furthermore, our method is compatible with various pretrained models. Compared to previous accelerating methods, our approach is capable of generating videos with higher quality and resolution, *i.e.*, 5-seconds, 720×1280, 24fps.

## 1 Introduction

Video generation has garnered significant attention due to its ability to simulate the real physical world (Agarwal et al., 2025; OpenAI, 2024), as well as its promising applications in entertainment, such as content creation (Ma et al., 2025; Wang et al., 2024c; Kong et al., 2024; Yang et al., 2025; Chen et al., 2024a; Zheng et al., 2024; Guo et al., 2023; Jin et al., 2025; Blattmann et al., 2023; Zhang et al., 2024), filmmaking (Kuaishou, 2024; OpenAI, 2024), video games (Yang et al., 2024; Valevski et al., 2024), and customized media generation (Ma et al., 2024; Hu et al., 2024; Lin et al., 2025a; Kuaishou, 2024; Xing et al., 2025).

With advancements in data curation pipeline (Kong et al., 2024; Agarwal et al., 2025) and scalable model architectures (Peebles & Xie, 2023), diffusion models (Karras et al., 2022; Ho et al., 2020) and flow matching (Liu et al., 2023a; Lipman et al., 2023) have emerged as widely used frameworks in video generation, owing to their impressive generative capabilities. However, video diffusion models require iterative denoising of Gaussian noise to generate the final videos, making them typically demand dozens of inference steps. This process is both slow and computationally intensive. For instance, HunyuanVideo (Kong et al., 2024) requires 3234s to generate a 5s, 720×1280, 24fps video on a single NVIDIA A100 GPU, as shown in Fig. 1.

Recently, significant progress has been made in the field of accelerating image diffusion models by distillation (Salimans & Ho, 2022; Frans et al., 2025; Berthelot et al., 2023; Yan et al., 2024; Yin et al., 2024b; Luo et al., 2024; Sauer et al., 2024b; Wang et al., 2023b; Xu et al., 2024; Lin et al., 2024; Sauer et al., 2024a; Yin et al., 2024a). However, distilling video diffusion models remains a challenge that requires further exploration. Although distillation methods for image diffusion models can be extended to video diffusion models, they require a large amount of data and computational resources (Lin et al., 2025b; Yin et al., 2024c) due

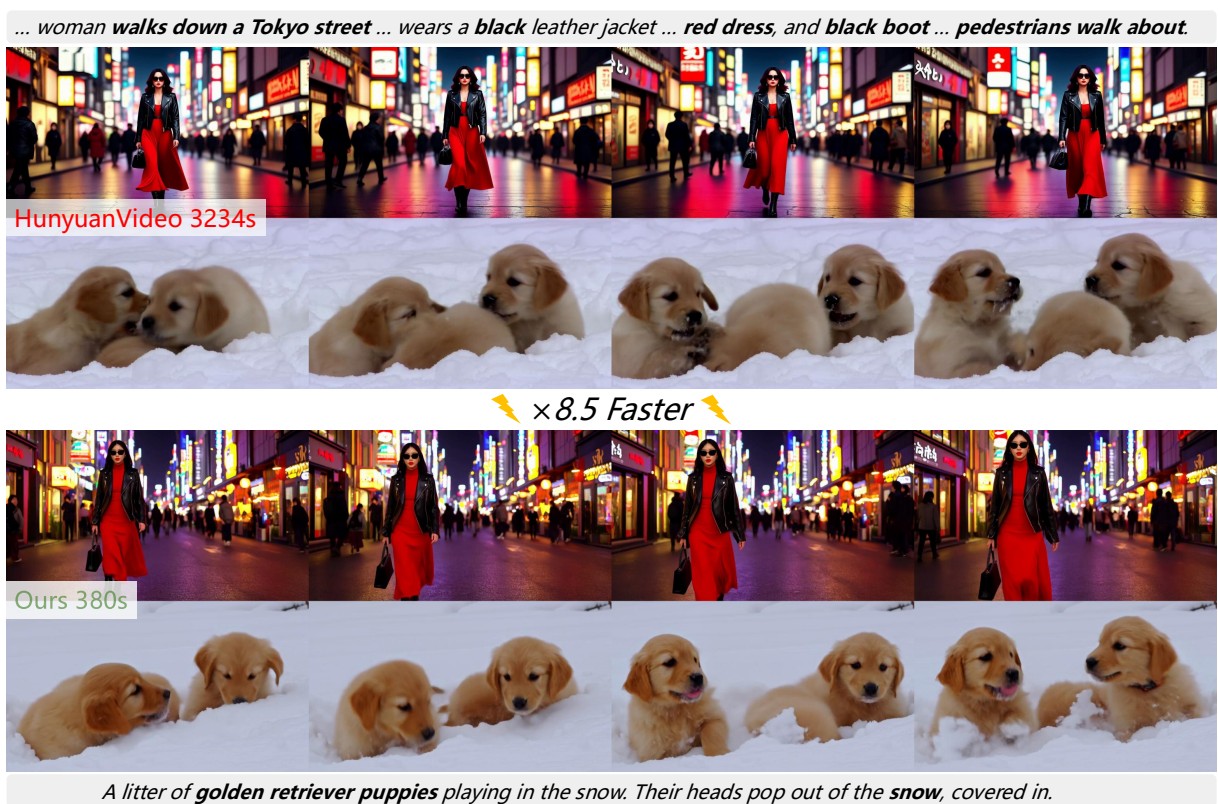

Figure 1: Video diffusion models can generate high-quality videos, but they require dozens of inference steps, resulting in slow generation process. For instance, HunyuanVideo (Kong et al., 2024) takes 3234 seconds to generate a 5-seconds, 720×1280, 24fps video on a single A100 GPU. In contrast, our method accelerates video diffusion models through distillation, achieving 8.5× improvement in generation speed while maintaining comparable quality.

to the use of useless data points, which do not lie on the denoising trajectories of the teacher model. These useless data points cause the teacher model to offer unreliable guidance for the student model, adversely affecting the video quality. To be concrete, the useless data points during distillation are caused by dataset mismatching or Gaussian noise mismatching. Dataset mismatching refers to the inconsistency between the dataset used for training the teacher model and the dataset utilized during the distillation process, which arises due to the difficulty in accessing the training dataset of the teacher model. Gaussian noise mismatching refers to the misalignment between Gaussian noise and the data, a phenomenon that occurs during the forward diffusion operation, also called the flow operation.

Moreover, videos exhibit complex spatio-temporal relationships, making them difficult to model. As a result, video diffusion models typically involve billions of parameters and high input dimensionality. For example, HunyuanVideo contains 13 billion parameters, and a 5s, 720×1280, 24fps video is represented by more than 100K tokens. This further complicates the realization of efficient video diffusion distillation.

In this paper, we begin with a detailed analysis of the challenges present in existing diffusion distillation models. Based on the analysis, we try to avoid the use of useless data points during the distillation process and propose a novel efficient distillation method, namely **AccVideo**, which aims to accelerate video diffusion models with synthetic dataset. Specifically, we first present a synthetic dataset, SynVid, comprising 110K denoising trajectories and high-quality videos generated by the teacher model (Kong et al., 2024) with fine-grained text prompts. The data points on the denoising trajectories are intermediate results leading to the correct output, making them all valid and meaningful. Then, we design a trajectory-based few-step guidance that selects a few of data points from each denoising trajectory to construct a shorter noise-to-

video mapping path, enabling the student model to generate videos in fewer steps. To further exploit the data distribution captured by our synthetic dataset, we propose an adversarial training strategy to align the intermediate output distribution of the student model with that of our synthetic dataset at each diffusion timestep, thereby enhancing the quality of generated videos. We obviate complex regularization designs for the adversarial training (Lin et al., 2025b) by combining the trajectory-based few-step guidance. It is noteworthy that the data points used in our method deliver precise guidance for the student model, markedly enhancing training efficiency and reducing the number of data. Our model is trained using only 8 A100 GPUs with 38.4K synthetic data for 12 days, yet it is capable of generating high-quality 5s, 720×1280, 24fps videos. We validate the effectiveness of our method across different teacher models (Kong et al., 2024; Wan et al., 2025) and generation tasks, including text-to-video (T2V) and image-to-video (I2V).

To summarize, our contributions are as follows:

- We provide a detailed analysis of existing diffusion distillation models and propose a novel efficient method to accelerate video diffusion models, eliminating the presence of useless data points and enabling efficient distillation.

- We present SynVid, a synthetic video dataset containing 110K high-quality synthetic videos, denoising trajectories, and corresponding fine-grained text prompts.

- We design a trajectory-based few-step guidance that leverages key data points from the synthetic dataset to learn the noise-to-video mapping with fewer steps and an adversarial training strategy, which effectively utilizes the distribution information captured by SynVid, thereby enhancing the performance.

- Extensive experiments demonstrate that we achieve $8.5\times$ improvement in generation speed compared to the teacher model (Kong et al., 2024) while maintaining comparable performance. Moreover, we produce videos with higher quality and resolution, *i.e.*, 5-seconds, 720×1280, 24fps, compared to previous accelerating methods.

## 2 Related Work

**Video Diffusion Models.** With the advancement of large-scale video data and models, video diffusion models have achieved remarkable success. The pioneering work VDM (Ho et al., 2022b) extends the 2D U-Net (Ronneberger et al., 2015) used in image generation to 3D UNet and proposes joint training with both images and videos. He et al. (2022); Zhou et al. (2022); Wang et al. (2023a); Guo et al. (2023); Chen et al. (2024a) adopt the latent diffusion model (LDM) (Rombach et al., 2022) to learn video distribution in the latent space, significantly reducing computational complexity. To further increase the resolution of generated videos, Blattmann et al. (2023); Ho et al. (2022a); Wang et al. (2024c) employ cascaded video diffusion models, which decomposes the generation process into subtasks, *i.e.*, key frame generation, frame interpolation, and super-resolution. With the impressive video generation capabilities shown by Sora (OpenAI, 2024), the diffusion transformer architecture (DiT) (Peebles & Xie, 2023) has gradually become the mainstream backbone for video diffusion models. Latte (Ma et al., 2025) and GenTron (Chen et al., 2024b) explore different variants of DiT for video generation. Snap Video (Menapace et al., 2024), HunyuanVideo (Kong et al., 2024), Wan (Wan et al., 2025), and Cosmos (Agarwal et al., 2025) replace 1D+2D self-attention with 3D self-attention and leverage more careful data curation pipelines, larger-scale models, and more computational resources, achieving state-of-the-art opensource video generation performance. Hu et al. (2025); Guo et al. (2024) attempt to achieve image-to-video generation using motion representations and adapters. For more details on video diffusion models, please refer to Wang et al. (2025) and Xing et al. (2024).

**Accelerating Image Diffusion Models.** Here, we focus on distillation techniques to accelerate image diffusion models. Salimans & Ho (2022); Frans et al. (2025); Berthelot et al. (2023) progressively distill the teacher model, enabling the student model to learn the noise-to-image mapping with fewer inference steps. PeRFlow (Yan et al., 2024) divides the denoising process into several windows and learns the data mapping within each window. LCM (Luo et al., 2023) utilize consistency models (Song et al., 2023) in the image latent space to accelerate pretrained image diffusion models. Inspired by Variation Score Distillation

(VSD) (Wang et al., 2024d) and Score Distillation Sampling (SDS) (Poole et al., 2023), Yin et al. (2024b); Luo et al. (2024); Yin et al. (2024a); Sauer et al. (2024b) propose the distribution matching loss, which aligns the real and fake image distributions by utilizing the score function derived from the diffusion model. Additionally, Wang et al. (2023b); Xu et al. (2024); Lin et al. (2024); Sauer et al. (2024a) employ adversarial loss to train few-step or one-step image generators, using diffusion models as the feature extractors. Liu et al. (2023b); Luhman & Luhman (2021) share similarities with our approach, they also utilize synthetic data to learn the noise-to-data mapping in fewer steps. However, they overlook the distribution information contained in the denoising trajectory, resulting blurry outputs.

**Accelerating Video Diffusion Models.** Recent studies primarily accelerate video diffusion models from four perspectives: attention mechanism acceleration, efficient model architectures, high-compression-rate Variational Autoencoders (VAEs) (Kingma, 2014), and distillation techniques. Zhang et al. (2025b); Xi et al. (2025); Zhang et al. (2025a); Liu et al. (2025b); Zhao et al. (2025); Lv et al. (2025) accelerate video generation by reducing redundant computations within the attention mechanism. LinGen (Wang et al., 2024b) employs the linear-complexity Mamba2 block (Dao & Gu, 2024), significantly reducing the training computational costs. LTX-Video (HaCohen et al., 2024) utilizes a carefully designed Video-VAE that achieves a high compression ratio, reducing the input dimensions of the diffusion model and enhancing generation speed. AnimateDiff-Lightning (Lin & Yang, 2024) extends progressive adversarial distillation (Lin et al., 2024) to video diffusion models. T2V-Turbo (Li et al., 2024) and T2V-Turbo-V2 (Li et al., 2025) leverage refined consistency distillation loss (Luo et al., 2023) to reduce the number of inference steps and enhance video quality using reward models. Concurrent works, CausVid (Yin et al., 2024c) and APT (Lin et al., 2025b), leverage distribution matching loss and adversarial loss, respectively, to distill video diffusion models. Although these methods also employ trajectory-based distillation, they typically rely on real data combined with forward diffusion to obtain the start points as detailed in Sec. 3.2, which inevitably introduces useless data points and causes the teacher model to provide unreliable guidance. In contrast, our method leverages synthetic data to eliminate these useless data points during distillation. Moreover, we further explore the rich prior encoded in intermediate denoising trajectories, which further improves the performance, efficiently achieving the generation of higher-quality and higher-resolution videos, *i.e.*, 5s, 720×1280, 24fps. For more details on efficient diffusion models, please refer to Shen et al. (2025).

## 3 Preliminaries

### 3.1 Flow Matching

In this section, we provide a brief overview about Flow Matching (Lipman et al., 2023), which is commonly used in generative tasks. Flow Matching transforms a complex data distribution $p_0(x)$ into the simple standard normal distribution $p_1(x) = \mathcal{N}(0, I)$ through a conditional probability paths $p_t(x|x_0)$, where $x_0 \sim p_0(x)$ and $t \in [0, 1]$. In Lipman et al. (2023), the conditional probability paths have the form:

$$p_t(x|x_0) = \mathcal{N}(x \mid \mu_t(x_0), \sigma_t(x_0)^2 I), \tag{1}$$

where $\mu_t(x_0)$ is the time-dependent mean, while $\sigma_t(x_0)$ describes the time-dependent scalar standard deviation. According to the Optimal Transport theory, the mean and the std are linearly changed in time,

$$\mu_t(x_0) = (1 - t)x_0, \sigma_t(x_0) = t. \tag{2}$$

Then, the sample $x_t = (1 - t)x_0 + tx_1 \sim p_t(x|x_0)$ is obtained through forward diffusion operation. Here, $x_1 \sim p_1(x)$. A model $v$ parameterized by $\theta$ is trained to predict the velocity $u_t(x|x_0) = x_1 - x_0$, which guides the sample $x_t$ towards the data $x_0$. The training loss has the form:

$$\mathcal{L}_{\text{FM}}(\theta) = \mathbb{E}_{t,p_0(x),p_1(x)} \left\| v_\theta(x_t, t) - u_t(x|x_0)) \right\|^2. \tag{3}$$

After training, a sampled Gaussian noise $x_1 \sim p_1(x)$ can be denoised to data $x_0 \sim p_0(x)$ by integrating the predicted velocity $v_\theta(x_t, t)$ through first-order Ordinary Differential Equation (ODE) solvers, such as the Euler method.

## 3.2 Analysis of Previous Distillation Methods

Previous diffusion distillation methods can be broadly categorized into two classes: (1) knowledge distillation (Salimans & Ho, 2022; Frans et al., 2025; Yan et al., 2024; Berthelot et al., 2023; Luo et al., 2023; Li et al., 2024; 2025), where student models are trained to mimic the denoising process of teacher models with fewer inference steps, *e.g.*, mapping $x_t$ to $x_{t'}$ or $x_0$ using one step, as shown in Fig. 2 a).

The start data points $x_t$ are obtained through the forward diffusion operation. However, these methods inadvertently distill useless start data points $x_t$, which do not lie on the denoising trajectories of the teacher model due to the dataset mismatching or Gaussian noise mismatching, as depicted in Fig. 2 b). When denoising such useless data points using the teacher model, it often produces inaccurate results, as highlighted by the red box in Fig. 2 b). Consequently, it can lead to unreliable guidance for the student model during distillation; (2) distribution distillation (Yin et al., 2024b; Luo et al., 2024; Yin et al., 2024a; Sauer et al., 2024a; Xu et al., 2024; Lin et al., 2025b), which leverages diffusion models to compute distribution matching loss or adversarial loss (Wang et al., 2023b) to optimize the student model, *e.g.*, feeding $x_t^{fake}$ and $x_t^{real}$ into the pretrained diffusion model to derive meaningful guidance, as illustrated in Fig. 2 c). However, they may still rely on useless data points $x_t^{fake}$ and $x_t^{real}$ due to the dataset mismatching or Gaussian noise mismatching, as shown in Fig. 2 d), which leads the diffusion model to provide inaccurate guidance.

We count the frequency of useless data points in relation to the degree of data mismatching and Gaussian noise mismatching in our 1D toy experiment. We define the degree of mismatching between the distillation dataset $p_d(x)$ and the training dataset $p(x)$ as follows,

$$M = \sum_{x_d \in p_d(x)} min\{|x_d - x| \text{ for } x \in p(x)\}. \quad (4)$$

Fig. 2 e) shows the results. Even without dataset mismatching, *i.e.*, $M = 0$, the presence of Gaussian noise mismatching still produces useless data points. As the degree of dataset mismatching $M$ increases, it becomes evident that the frequency of useless data points grows significantly. Additionally, we conduct

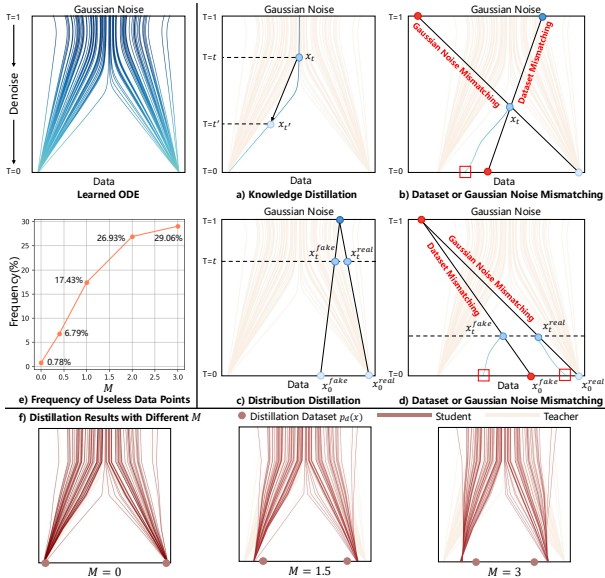

Figure 2: **1D Toy Experiment.** Top-Left illustrates the setup of our 1D toy experiment; Top-Right (a,b,c,d) illustrates the pipelines and problems of existing distillation methods; Middle (e) shows that useless data points exist in practice; Bottom row (f) demonstrates that useless data points during distillation degrade the performance. Specifically, **a)** illustrates the knowledge distillation methods, where a student model is trained to mimic the teacher model's denoising process. **b)** highlights the challenges posed by dataset or Gaussian noise mismatching in knowledge distillation, which can lead to unreliable guidance. **c)** demonstrates the distribution matching methods, which aims to align the output distribution of the student model with that of the teacher model. **d)** emphasizes the issue in distribution matching, which can result in inaccurate guidance. **e)** illustrates the frequency of useless data points in relation to $M$. **f)** shows the distillation results at various values of $M$.

distillation experiments followed by Yan et al. (2024) at different $M$. The results are illustrated in Fig. 2 f). As $M$ increases, more useless data points are used during distillation, making the teacher model provide incorrect guidance. Consequently, this leads the generated data to deviate from the training dataset $p(x)$ and the distillation dataset $p_d(x)$, demonstrating that useless data points are harmful to the distillation process.

## 4 Method

Our method aims to distill the pretrained video diffusion model (Kong et al., 2024) to reduce the number of inference steps, thereby accelerating video generation. Based on the anal-

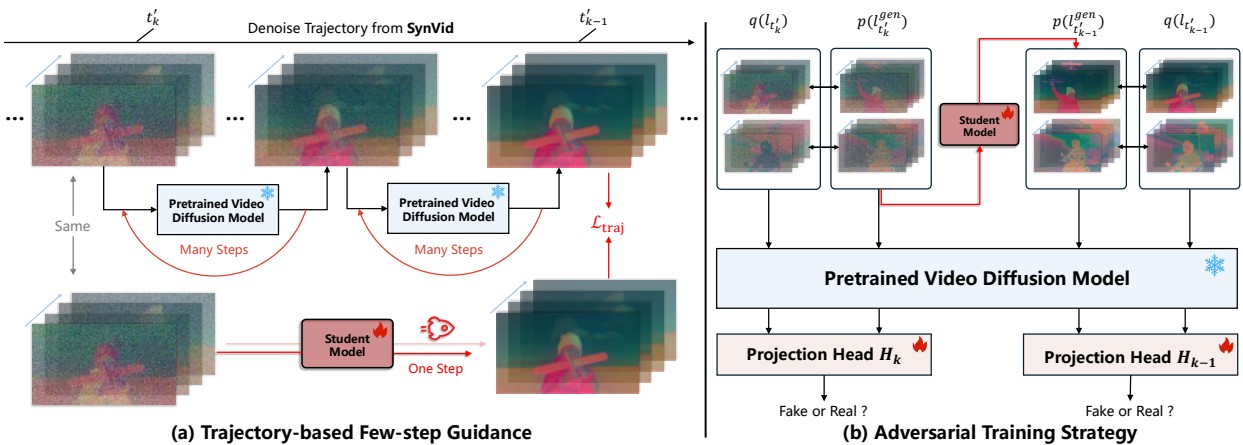

Figure 4: **Method Overview.** **(a)** Our method first designs a trajectory-based few-step guidance, which utilizes the key data points from the denoising trajectory to enable the student model to mimic the denoising process of the pretrained video diffusion model with fewer steps. **(b)** To fully exploit the data distribution at each diffusion timestep captured by our synthetic dataset, we propose an adversarial training strategy to align the intermediate output distribution of the student model with that captured by our synthetic dataset.

ysis in Sec. 3.2, we avoid using useless data points during the distillation process. Specifically, we first introduce a synthetic video dataset, SynVid, which leverages the teacher model to generate high-quality synthetic videos and denoising trajectories (Sec. 4.1). Subsequently, we propose a trajectory-based few-step guidance that selects key data points from the denoising trajectories

and learns the noise-to-video mapping based on these data points, enabling video generation with fewer inference steps (Sec. 4.2). Additionally, we introduce an adversarial training strategy that exploits the data distribution at each diffusion timestep captured by SynVid, further enhancing the model's performance (Sec. 4.3). Fig. 4 illustrates the overview of our method.

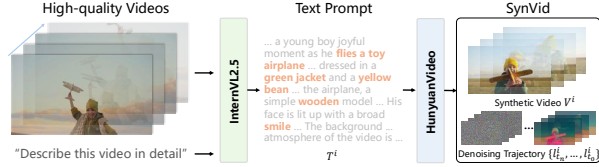

Figure 3: The pipeline of constructing SynVid.

## 4.1 SynVid: Synthetic Video Dataset

We employ HunyuanVideo (Kong et al., 2024) as our generator to produce synthetic data as shown in Fig. 3. HunyuanVideo $v_\theta$ is a text-to-video diffusion model utilizing the DiT architecture (Peebles & Xie, 2023), which operates in the latent space and is trained using the Flow Matching objective, as outlined in Eq. 3. We use the official code and settings[1] to generate our dataset.

Specifically, our synthetic dataset $\mathcal{D}_{\text{syn}} = \{V^i, l_{t_n}^i, l_{t_{n-1}}^i, ..., l_{t_0}^i, T^i | i \in [1, N]\}$ comprises high-quality synthetic videos $V_i$, denoising trajectories in latent space $\{l_{t_n}^i, l_{t_{n-1}}^i, ..., l_{t_0}^i | t_n = 1 > ... > t_0 = 0\}$, and their corresponding text prompts $T^i$, where $N = 110K$ represents the number of data, $n = 50$ denotes the number of inference steps, and $\{t_j | j \in [0, n]\}$ denote the inference diffusion

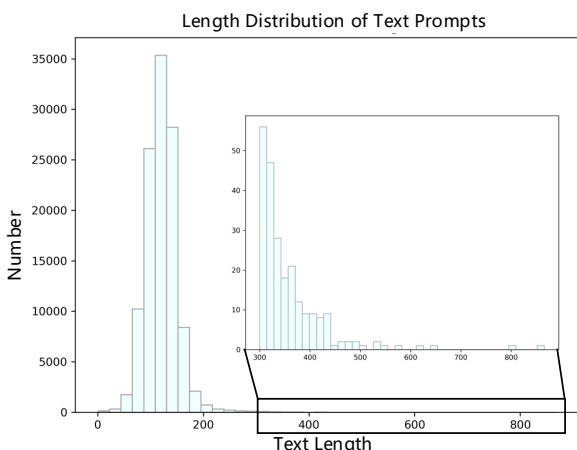

Figure 5: The length distribution of text prompts.

---

[1]https://github.com/Tencent/HunyuanVideo

timesteps. Recent methods (OpenAI, 2024; Betker et al., 2023) have demonstrated that fine-grained text prompts play a crucial role in video generation. Therefore, we leverage a multimodal large language model, *i.e.*, InternVL2.5-8B (Chen et al., 2024c), to annotate real videos and obtain high-quality text prompts. Fig. 5 shows the length distribution of text prompts, most text lengths are concentrated between 100 and 150, demonstrating that the text prompts provide rich informations, ensuring the generation quality. Moreover, these text prompts contain detailed descriptions across a wide range of scenarios, effectively enhancing the generalization ability of the student model. For a Gaussian noise $l_{t_n}^i$, the denoising trajectory is solved as follows,

$$l_{t_{j-1}}^i = l_{t_j}^i + (t_{j-1} - t_j) v_\theta(l_{t_j}^i, t_j, T^i), \tag{5}$$

the synthetic video $V^i$ can be decoded by VAE using the clean latent $l_{t_0}^i$. For brevity, we omit the text prompt input $T$ in following sections. It is noteworthy that we exclusively utilize the synthetic dataset $\mathcal{D}_{\text{syn}}$ during the distillation process. The data points in $\mathcal{D}_{\text{syn}}$ are intermediate results that lead to the correct output $l_{t_0}^i$, making them all valid and meaningful. This significantly aids in efficient distillation and reduces the data demand.

## 4.2 Trajectory-based Few-step Guidance

Video diffusion models typically require a large number of inference steps to generate videos, which is time-consuming and computationally intensive. To accelerate the generation process, we design a student model $s_\beta$ that utilizes denoising trajectories generated by the pretrained video diffusion model, *i.e.*, the teacher model, to learn the noise-to-video mapping with fewer inference steps. The student model shares the same architecture as the teacher model and is parameterized by $\beta$, which is initialized using the parameters $\theta$ of the teacher model.

Specifically, we select $m + 1$ key diffusion timesteps $\{t_m' = 1 > t_{m-1}' > ... > t_0' = 0\}$ and obtain their corresponding latents $\{l_{t_m'}^i, l_{t_{m-1}'}^i, ..., l_{t_0'}^i\}$ on each denoising trajectory. Then, we propose a trajectory-based loss to learn the denoising process of the teacher model with $m$ steps,

$$\mathcal{L}_{\text{traj}} = \mathbb{E}_{i,k} \left\| s_\beta(l_{t_{k+1}'}^i, t_{k+1}') - \frac{l_{t_k'}^i - l_{t_{k+1}'}^i}{t_k' - t_{k+1}'} \right\|^2, \tag{6}$$

where $k \in [0, m-1]$. By learning from these key latents, which construct a shorter path from Gaussian noise to video latent, our student model significantly reducing the number of inference steps. In our experiments, we set $m = 5$, which reduces the number of inference steps by a factor of 10 compared to the teacher model, greatly accelerating the generation process. Inspired by Fig. 3 in Liu et al. (2025a), we observed that for the teacher model, the outputs at adjacent diffusion timesteps become increasingly dissimilar as $t$ increases, making it more difficult to integrate. Therefore, we select $m = 5$ key diffusion timesteps as follows: $t_5' = 1.0$, $t_4' = 0.9651$, $t_3' = 0.9130$, $t_2' = 0.8235$, $t_1' = 0.6363$, and $t_0' = 0$, where each key diffusion timestep has a more similar learning difficulty.

## 4.3 Adversarial Training Strategy

In addition to the one-to-one mapping between Gaussian noise and video latents, our synthetic dataset also contains many latents $\{l_{t_k'}^i | i \in [1, N]\}$ at each diffusion timestep $t_k'$, which implicitly represent the data distribution $q(l_{t_k'})$. To fully unleash the knowledge in our synthetic dataset and enhance the performance of our student model $s_\beta$, we propose an adversarial training strategy to minimize adversarial divergence $D_{\text{adv}}(q(l_{t_k'}) \| p(l_{t_k'}^{\text{gen}}))$, where $p(l_{t_k'}^{\text{gen}})$ denotes the generated data distribution at diffusion timestep $t_k'$ of our student model $s_\beta$. The

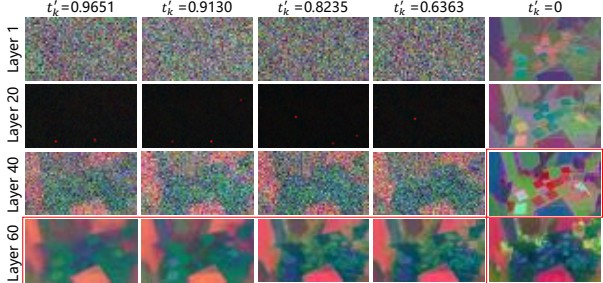

Figure 6: The illustration of features at different layers and diffusion timesteps of our feature extractor. The features within the red box are selected for discrimination.

training objective follows the standard Generative Adversarial Network (GAN) (Goodfellow et al., 2024):

$$\mathcal{L}_{\text{adv}} = \min_{\beta} \max_{\phi} \; \mathbb{E}_{k, q(l_{t'_k}), p(l^{\text{gen}}_{t'_k})} \big[ \log(D(l_{t'_k}, t'_k)) + \log(1 - D(l^{\text{gen}}_{t'_k}, t'_k)) \big], \tag{7}$$

where $D$ represents the discriminator parameterized by $\phi$. Since the discriminator needs to distinguish noisy latents at different diffusion timesteps, we design our discriminator $D$ to consist of a noise-aware feature extractor and $m$ timestep-aware projection heads $\{H_0, ..., H_{m-1}\}$,

$$D(l_{t'_k}, t'_k) = H_k\big(v_\theta(l_{t'_k}, t'_k)\big), \qquad D(l^{\text{gen}}_{t'_k}, t'_k) = H_k\big(v_\theta(l^{\text{gen}}_{t'_k}, t'_k)\big), \tag{8}$$

here, we leverage the frozen teacher model $v_\theta$ as the feature extractor inspired by Xu et al. (2024); Yin et al. (2024a); Lin et al. (2024). Fig. 6 shows the layer-wise behavior at different diffusion timesteps of $v_\theta$. When $t'_k > 0$, the deeper layer concentrate on capturing high-frequency details, so we select the output at the final layer, *i.e.*, the 60th layer for discrimination. When $t'_k = 0$, we choose the output at the 40th layer for discrimination, which contains fine-grained clues, such as sticky notes on the table. Each projection head $H_k$ is a 2D convolutional neural network used to output the label indicating where the latents $l_{t'_k}$ and $l^{\text{gen}}_{t'_k}$ comes from. The projection heads are timestep-aware, effectively handling noisy features at different diffusion timesteps and thereby facilitating adversarial learning. For latent $l^{\text{gen}}_{t'_k} \sim p_\beta(l^{\text{gen}}_{t'_k})$, it can be iteratively solved using $s_\beta$, similar to Eq. 5. However, it is time-consuming, which is not practical implementation. Therefore, we employ $m+1$ queues $\{Q_0, ..., Q_m\}$, which maintain the latents generated by $s_\beta$ at each diffusion timestep.

**Total Objective.** The training parameters include $\beta$ and the parameters of the projection heads. These parameters are trained using $\mathcal{L}_{\text{traj}}$ and $\lambda_{adv}\mathcal{L}_{\text{adv}}$, where $\lambda_{adv}$ is a hyperparameter. Algorithm 1 shows the distillation procedure.

**Discussion.** Although CausVid (Yin et al., 2024c) and APT (Lin et al., 2025b) also attempt to accelerate video diffusion models through distillation, they require training the teacher model on their own video datasets to mitigate dataset mismatching. However, it requires substantial computational resources, particularly for high-resolution video generation. Additionally, APT relies on complex regularization and few-step initialization to stabilize adversarial training. In contrast, the incorporation of our trajectory-based few-step guidance improves the stability of adversarial learning, eliminating complex training strategies. Compared to existing adversarial method (Xu et al., 2024; Yin et al., 2024a), our method avoids the forward diffusion process by leveraging intermediate data distributions, thereby preventing the distillation of uesless data points. This ensures that the teacher model provides more accurate and effective guidance to the student model.

---

**Algorithm 1** AccVideo Distillation Procedure

**Input** : Teacher model $v_\theta$, SynVid $\mathcal{D}_{\text{syn}}$, $m$.
**Output:** Distilled $m$-step student model $s_\beta$, projection heads $\{H_0, ..., H_{m-1}\}$.
   // Initialize student from teacher
1  $\beta \leftarrow \theta$
   // Select $m$ key timesteps on denoising trajectory
2  $t'_m = 1 > t'_{m-1} > ... > t'_0 = 0$
   // Initialize latent queues with $\varnothing$
3  $Q_0, ..., Q_m \leftarrow \varnothing$
4  **while** *train* **do**
5     **for** $k \leftarrow m-1$ **to** *0* **do**
         // Sample key data points
6        $\{l_{t'_m}, l_{t'_{m-1}}, ..., l_{t'_0}\} \sim \mathcal{D}_{\text{syn}}$
7
         // Update student model with $\mathcal{L}_{\text{traj}}$
8        $\mathcal{L}_{\text{traj}} \leftarrow \text{traj}(s_\beta, l_{t'_k}, l_{t'_{k+1}})$    // Eq. 6
9        $\beta \leftarrow \text{Update}(\beta, \mathcal{L}_{\text{traj}})$
10      Sample $z \sim \mathcal{N}(0, \mathbf{I})$
11      $Q_m.\text{push}(z, l_{t'_{m-1}}, ..., l_{t'_0})$ **if** $k = m-1$
12
         // Compute latent $l^{\text{gen}}_{t'_k} \sim p_\beta(l^{\text{gen}}_{t'_k})$
13      $l^{\text{gen}}_{t'_{k+1}}, l_{t'_{m-1}}, ..., l_{t'_0} \leftarrow Q_{k+1}.\text{pop}()$
14      $l^{\text{gen}}_{t'_k} \leftarrow \text{Solver}(s_\beta, l^{\text{gen}}_{t'_{k+1}}, t'_{k+1}, t'_k)$ // Eq. 5
15      $Q_k.\text{push}(l^{\text{gen}}_{t'_k}, l_{t'_{m-1}}, ..., l_{t'_0})$
16
         // Update student model and $H_k$ with $\mathcal{L}_{\text{adv}}$
17      $\mathcal{L}_{\text{adv}} \leftarrow \text{adv}(l^{\text{gen}}_{t'_k}, l_{t'_k}, t'_k, H_k, v_\theta)$    // Eq. 7
18      $\beta \leftarrow \text{Update}(\beta, \lambda_{adv}\mathcal{L}_{\text{adv}})$
         $H_k \leftarrow \text{Update}(H_k, \lambda_{adv}\mathcal{L}_{\text{adv}})$

---

## 5 Experiments

**Implementation Details.** Our student model and teacher model adopt the HunyuanVideo architecture (Kong et al., 2024) and are initialized using the officially released checkpoint. The teacher model remains frozen. The student model and projection heads are trained for 1200 iterations using the AdamW optimizer (Loshchilov & Hutter, 2019) with a learning rate of $5 \times 10^{-6}$ for 12 days, utilizing 8 A100 GPUs

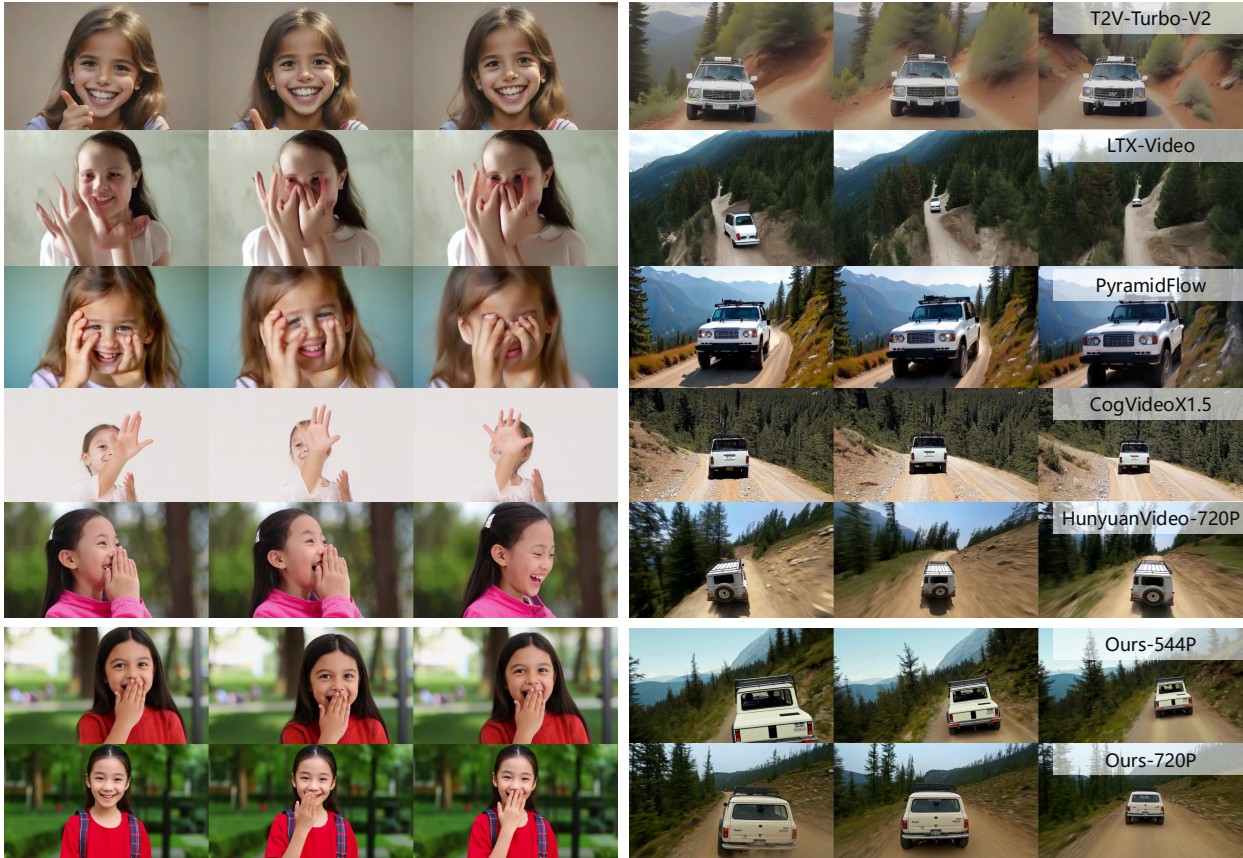

Figure 7: Qualitative results on text-to-video. **Left:** *a girl raises her left hand to cover her smiling mouth.* **Right:** *the camera follows behind a white vintage SUV with a black roof rack as it speeds up a steep dirt road surrounded by pine trees on a steep mountain slope.*

and gradient accumulation to achieve a total batch size of 32. The hyperparameter $\lambda_{\mathrm{adv}}$ is set to 0.1. We use SynVid as described in Sec. 4.1 as our distillation dataset, where the resolution of the latents and videos is $68 \times 120 \times 24$ and $544 \times 960 \times 93$, respectively. Generating the full SynVid dataset takes approximately 7K GPU-hours on A100 GPUs, where we leverage sequence parallel to accelerate the generation. After distillation, we can generate higher-resolution videos, i.e., $720 \times 1280 \times 129$. The inference process in our experiments is conducted on a single A100 GPU.

**Evaluation.** Our method is evaluated on VBench (Huang et al., 2024), a comprehensive benchmark that is commonly used to evaluate the video quality. Specifically, VBench evaluates video generation models across 16 dimensions using 946 prompts. Each prompt is sampled five times to reduce randomness.

**Baselines.** The baselines used in our experiments include VideoCrafter2 (Chen et al., 2024a), which is a 3DUNet-based model, as well as OpenSora (Zheng et al., 2024), HunyuanVideo (Kong et al., 2024), and CogVideoX (Yang et al., 2025), which are DiT-based models. Additionally, we compare with faster models such as T2V-Turbo-V2 (Li et al., 2025), which utilize distillation to accelerate video generation, and LTX-Video (HaCohen et al., 2024), which employs a high-compression VAE, as well as PyramidFlow (Jin et al., 2025), which combines autoregressive and diffusion models for efficient video generation.

## 5.1 Comparisons on Text-to-Video Generation

**Quantitative Evaluation.** The evaluations on VBench are presented in Table 1. T2V-Turbo-V2 (Li et al., 2025) achieves leading results in total score, primarily due to their highest performance in dynamic degree (improving by at least 13%). However, this may be attributed to its imperfect temporal consistency in local areas, as evidenced by its poor performance in temporal flickering. Moreover, it struggles to generate

Table 1: Text-to-video comparisons on VBench Huang et al. (2024). The results marked with ∗ are tested by us, while the other results are obtained from the official VBench leaderboard.

| Resolution | Method | H × W × L | Total Score | Quality Score | Semantic Score | Subject Consistency | Background Consistency | Temporal Flickering | Motion Smoothness | Dynamic Degree | Aesthetic Quality |
|---|---|---|---|---|---|---|---|---|---|---|---|
| Low | VideoCrafter2 | 320 × 512 × 16 | 80.44% | 82.20% | 73.42% | 96.85% | 98.22% | 98.41% | 97.73% | 42.50% | 63.13% |
| | T2V-Turbo-V2 | 320 × 512 × 16 | 83.52% | 85.13% | 77.12% | 95.50% | 96.71% | 97.35% | 97.07% | 90.00% | 62.61% |
| Medium | LTX-Video | 512 × 768 × 121 | 80.00% | 82.30% | 70.79% | 96.56% | 97.20% | 99.34% | 98.96% | 54.35% | 59.81% |
| | CogVideoX | 480 × 720 × 49 | 81.61% | 82.75% | 77.04% | 96.23% | 96.52% | 98.66% | 96.92% | 70.97% | 61.98% |
| | HunyuanVideo-544P∗ | 544 × 960 × 93 | 82.67% | 84.43% | 75.59% | 94.33% | 97.13% | 99.03% | 98.64% | 76.38% | 61.97% |
| | Ours-544P∗ | 544 × 960 × 93 | 83.26% | 84.58% | 77.96% | 94.46% | 97.45% | 99.18% | 98.79% | 75.00% | 62.08% |
| High | PyramidFlow | 768 × 1280 × 121 | 81.72% | 84.74% | 69.62% | 96.95% | 98.06% | 99.49% | 99.12% | 64.63% | 63.26% |
| | OpenSora V1.2 | 720 × 1280 × 204 | 79.76% | 81.35% | 73.39% | 96.75% | 97.61% | 99.53% | 98.50% | 42.39% | 56.85% |
| | CogVideoX1.5 | 768 × 1360 × 81 | 82.17% | 82.78% | 79.76% | 96.87% | 97.35% | 98.88% | 98.31% | 50.93% | 62.79% |
| | HunyuanVideo-720P | 720 × 1280 × 129 | 83.24% | 85.09% | 75.82% | 97.37% | 97.76% | 99.44% | 98.99% | 70.83% | 60.36% |
| | Ours-720P (distilled at 544P)∗ | 720 × 1280 × 129 | 82.77% | 84.38% | 76.34% | 94.20% | 96.87% | 99.01% | 98.90% | 75.00% | 60.61% |
| | Ours-720P (distilled at 720P)∗ | 720 × 1280 × 129 | 83.21% | 84.94% | 76.30% | 97.32% | 97.83% | 99.34% | 98.93% | 70.27% | 60.86% |

| Resolution | Method | H × W × L | Imaging Quality | Object Class | Multiple Objects | Human Action | Color | Spatial Relationship | Scene | Appearance Style | Temporal Style | Overall Consistency |
|---|---|---|---|---|---|---|---|---|---|---|---|---|
| Low | VideoCrafter2 | 320 × 512 × 16 | 67.22% | 92.55% | 40.66% | 95.00% | 92.92% | 35.86% | 55.29% | 25.13% | 25.84% | 28.23% |
| | T2V-Turbo-V2 | 320 × 512 × 16 | 71.78% | 95.33% | 61.49% | 96.20% | 92.53% | 43.32% | 56.40% | 24.17% | 27.06% | 28.26% |
| Medium | LTX-Video | 512 × 768 × 121 | 60.28% | 83.45% | 45.43% | 92.80% | 81.45% | 65.43% | 51.07% | 21.47% | 22.62% | 25.19% |
| | CogVideoX | 480 × 720 × 49 | 62.90% | 85.23% | 62.11% | 99.40% | 82.81% | 66.35% | 53.20% | 24.91% | 25.38% | 27.59% |
| | HunyuanVideo-544P∗ | 544 × 960 × 93 | 65.57% | 88.67% | 67.69% | 94.80% | 92.07% | 67.39% | 51.23% | 19.43% | 23.96% | 26.80% |
| | Ours-544P∗ | 544 × 960 × 93 | 65.64% | 92.99% | 67.33% | 95.60% | 94.11% | 75.70% | 54.72% | 19.87% | 23.71% | 27.21% |
| High | PyramidFlow | 768 × 1280 × 121 | 65.01% | 86.67% | 50.71% | 85.60% | 82.87% | 59.53% | 43.20% | 20.91% | 23.09% | 26.23% |
| | OpenSora V1.2 | 720 × 1280 × 204 | 63.34% | 82.22% | 51.83% | 91.20% | 90.08% | 68.56% | 42.44% | 23.95% | 24.54% | 26.85% |
| | CogVideoX1.5 | 768 × 1360 × 81 | 65.02% | 87.47% | 69.65% | 97.20% | 87.55% | 80.25% | 52.91% | 24.89% | 25.19% | 27.30% |
| | HunyuanVideo-720P | 720 × 1280 × 129 | 67.56% | 86.10% | 68.55% | 94.40% | 91.60% | 68.68% | 53.88% | 19.80% | 23.89% | 26.44% |
| | Ours-720P (distilled at 544P)∗ | 720 × 1280 × 129 | 66.70% | 89.71% | 66.35% | 94.00% | 94.61% | 71.74% | 50.75% | 20.16% | 23.44% | 26.92% |
| | Ours-720P (distilled at 720P)∗ | 720 × 1280 × 129 | 66.83% | 87.93% | 67.21% | 94.60% | 93.48% | 71.26% | 52.64% | 20.22% | 23.78% | 26.27% |

high-resolution videos, whereas our model retains this ability. In addition to T2V-Turbo-V2 (Li et al., 2025) and our teacher model, HunyuanVideo (Kong et al., 2024), our method surpasses all the other compared baselines at the same resolution level. Compared to our teacher model, we achieve better performance at medium resolution. When directly extrapolating the 544P-distilled model to 720P, the Total Score decreases from 83.24% to 82.77% due to the resolution mismatch between the distillation and evaluation resolutions. However, when distill at 720P dataset, the Total Score improves to 83.21%, which is comparable to the teacher model. In particular, we achieve exceptional performance in color and spatial relationship, indicating that our model can effectively interpret text prompts and generate videos coherent with the text. This not only demonstrates the effectiveness of our method but also highlights that our synthetic dataset contains high-quality text prompts and data points.

**Qualitative Evaluation.** Fig. 7 presents the qualitative results. Compared to T2V-Turbo (Li et al., 2025), LTX-Video (HaCohen et al., 2024), and Pyramid-Flow (Jin et al., 2025), our method generates videos with fewer artifacts, such as the hands of the little girl. Compared to CogVideoX1.5 (Yang et al., 2025), we produce higher-fidelity videos with better backgrounds. Compared to HunyuanVideo (Kong et al., 2024), our method better aligns with the text prompts, such as the vintage SUV.

Table 2: Comparison of inference time. Here, we use a single A100 GPU to measure the time required for generating a video, which includes text encoding, VAE decoding, and diffusion time.

| Resolution | Method | Time(s) |
|---|---|---|
| Medium | CogVideoX-5B | 219 |
| | HunyuanVideo-544P | 704 |
| | Ours-544P | **91** |
| High | CogVideoX1.5-5B | 926 |
| | HunyuanVideo-720P | 3234 |
| | Ours-720P | **380** |

**Inference Time.** Table 2 presents the comparison of the inference time. Our model requires only 5 inference steps to generate videos, it achieves a 7.7-8.5× improvement in generation speed compared to the teacher model. Even compared to models with half the model size of ours, *e.g.*, CogVideoX-5B and CogVideoX1.5-5B (Yang et al., 2025), our model is still 2.7× faster.

## 5.2 Ablation Studies

The following ablation studies are conducted at medium resolution. The quantitative results on VBench are reported in Table 3.

**Trajectory-based Few-step Guidance.** The trajectory-based few-step guidance is designed to guide the student model to learn the denoising trajectories of the teacher model with fewer steps. As illustrated in Fig. 8 (a) and (b), we observe that the teacher model struggles to generate clear videos with fewer inference

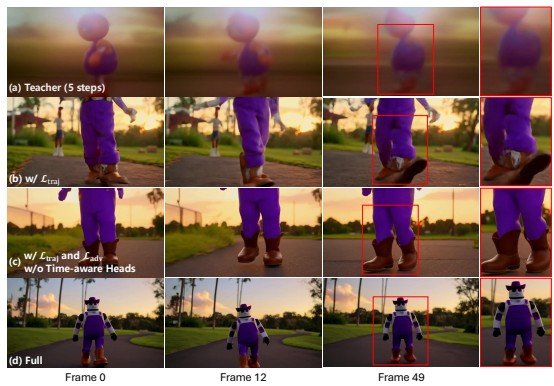

Figure 8: Qualitative ablation study results. Please zoom in for details.

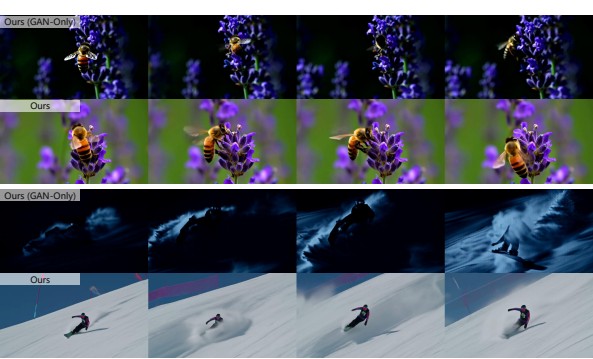

Figure 9: Ablation study on training with adversarial loss alone.

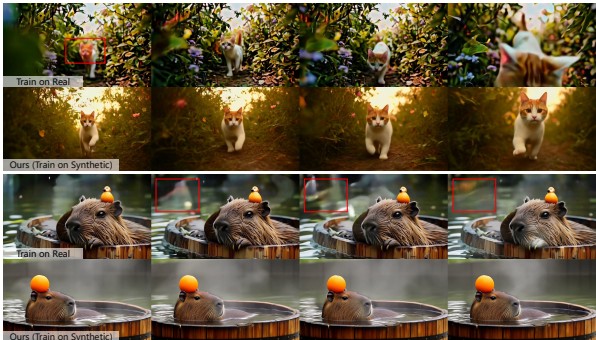

Figure 10: Ablation study on distillation dataset. When using a real dataset for distillation, the generated videos tend to be blurry (**Top**) or exhibit unnatural backgrounds (**Bottom**).

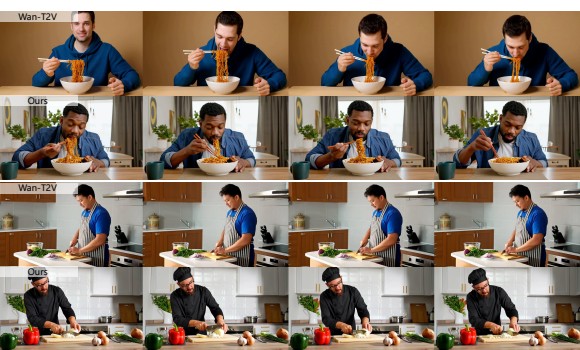

Figure 11: Distillation based on Wan-T2V-14B. Our method can be successfully applied to various teacher models while preserving their generative capacity, even in more challenging scenarios.

steps, while distilling with our trajectory-based loss is capable of producing clearer videos in just 5 steps. We greatly accelerate video generation by reducing the number of inference steps.

**Adversarial Training Strategy.** As shown in Fig. 8 (b) and (c), although the trajectory-based few-step guidance accelerates video generation, the generated videos contain artifacts with unnatural motions. In contrast, when the adversarial training strategy is employed, we can generate higher-quality videos. This further demonstrates its effectiveness. However, training with adversarial loss alone leads to unstable optimization and mode collapse as shown in Fig. 9. Our trajectory-based few-step guidance effectively stabilizes adversarial training and eliminates complex designs. Thus, trajectory-based few-step guidance and adversarial loss are both indispensable.

Table 3: **Ablation Studies.** Quantitative results on VBench.

| Method | Total Score | Quality Score | Semantic Score |
|---|---|---|---|
| w/$\mathcal{L}_{\text{traj}}$ | 82.56% | 83.88% | 77.28% |
| w/$\mathcal{L}_{\text{traj}}$ and $\mathcal{L}_{\text{adv}}$ | 83.08% | 84.37% | 77.91% |
| Real data + PCM | 79.52% | 82.26% | 68.55% |
| Real data + DMD | 82.95% | 84.23% | 77.86% |
| Full | **83.26%** | **84.58%** | **77.96%** |

**Timestep-Aware Projection Heads.** We conduct an ablation study, which only uses one projection head to discriminate the extracted timestep-aware features. As illustrated in Fig. 8 (c) and (d), the generated videos better align with the text prompts and produce more coherent videos when using our timestep-aware projection heads. We hypothesize that when a single projection head is used, it may struggle to distinguish features from different diffusion timesteps, *i.e.*, different noise levels. In contrast, the timestep-aware projection heads can more effectively learn variance present in these timestep-aware features.

**Distillation Datasets and Methods.** To evaluate the impact of synthetic and real datasets on distillation, we compared our method with FastVideo (hao-ai lab, 2025), which is distilled using real datasets[2] based on PCM (Wang et al., 2024a). Specifically, both methods use same teacher model. As shown in Fig. 10 and Table 3, although FastVideo employs more computational resources, *i.e.*, 64 H100, its results are blurry and exhibit unnatural background. In contrast, our method leverages synthetic dataset to distill, which is more efficient and can produce better videos, further demonstrating the effectiveness of synthetic data. To further evaluate our method, we also compare with FastVideo using real data and DMD as the distillation objective, with the same HunyuanVideo teacher model and identical inference steps (5 steps). As shown in Table 3, training with real data and DMD leads to inferior performance compared to our method, demonstrating the importance of leveraging synthetic datasets and our proposed distillation method.

**Timestep Selection.** To justify the choice of our timestep schedule, we conduct an ablation comparing our schedule with a uniform timestep grid ($t = \{1000.0, 810.9, 605.7, 378.3, 225.8, 0.0\}$) selected from our synthetic trajectories. The results on VBench are reported in Table 4. Our timestep schedule outperforms the uniform grid, validating the design choice empirically. We hypothesize that the non-uniform schedule, which allocates more timesteps in the high-noise region, better accommodates the varying learning difficulty across diffusion timesteps.

Table 4: Comparison of timestep schedules

| Timestep Schedule | Total Score | Quality Score | Semantic Score |
|---|---|---|---|
| Uniform grid-544P | 82.86% | 84.33% | 76.98% |
| Ours-544P | **83.26%** | **84.58%** | **77.96%** |

### 5.3 More Applications

**Different Teacher Models.** Our proposed method is applicable to a variety of teacher models, such as Wan (Wan et al., 2025). Here, we apply our distillation method to Wan-T2V-14B[3], successfully reducing the number of function evaluations (NFE) from 100 to 10, achieving a 9.6× speed-up. As shown in Fig. 11, our method effectively preserves the generative capacity of the teacher model, even in challenging scenarios such as eating and cutting.

**Image-to-Video.** In addition to the T2V task, our method can also be applied to the I2V task. Specifically, we distill based on the Wan-I2V-14B[4] and successfully reduce NFE from 80 to 10, achieving a 6.8× speed-up. As shown in Fig. 12, our method consistently maintains high generation quality.

## 6 Conclusion

In this paper, we first analyze the challenges faced by existing diffusion distillation methods, which are caused by the use of useless data points. Based on this insight, we propose a novel efficient distillation method to accelerate video diffusion models. Specifically, we first construct a synthetic video dataset, SynVid, which contains valid and meaningful data points for distillation. Then, we introduce a trajectory-based few-step guidance that enables the student model to generate videos in just 5 steps, significantly accelerating the generation speed compared to the teacher model. To further enhance the video quality, we design an adversarial training strategy that leverages the data distribution captured by our synthetic video dataset. Our model achieves 8.5× improvements in generation speed compared to the teacher model while maintaining comparable performance. Moreover, our method can be applied to various pretrained video diffusion models and generation tasks.

### Broader Impact Statement

Our method significantly reduces the computational cost of high-quality video generation, which has positive applications for content creation, entertainment, and education. However, the ability to efficiently generate realistic videos could potentially be misused for generating deepfakes or other misleading content. To mitigate

---

[2] `https://huggingface.co/datasets/LanguageBind/Open-Sora-Plan-v1.1.0`
[3] `https://huggingface.co/Wan-AI/Wan2.1-T2V-14B`
[4] `https://huggingface.co/Wan-AI/Wan2.1-I2V-14B-480P-Diffusers`

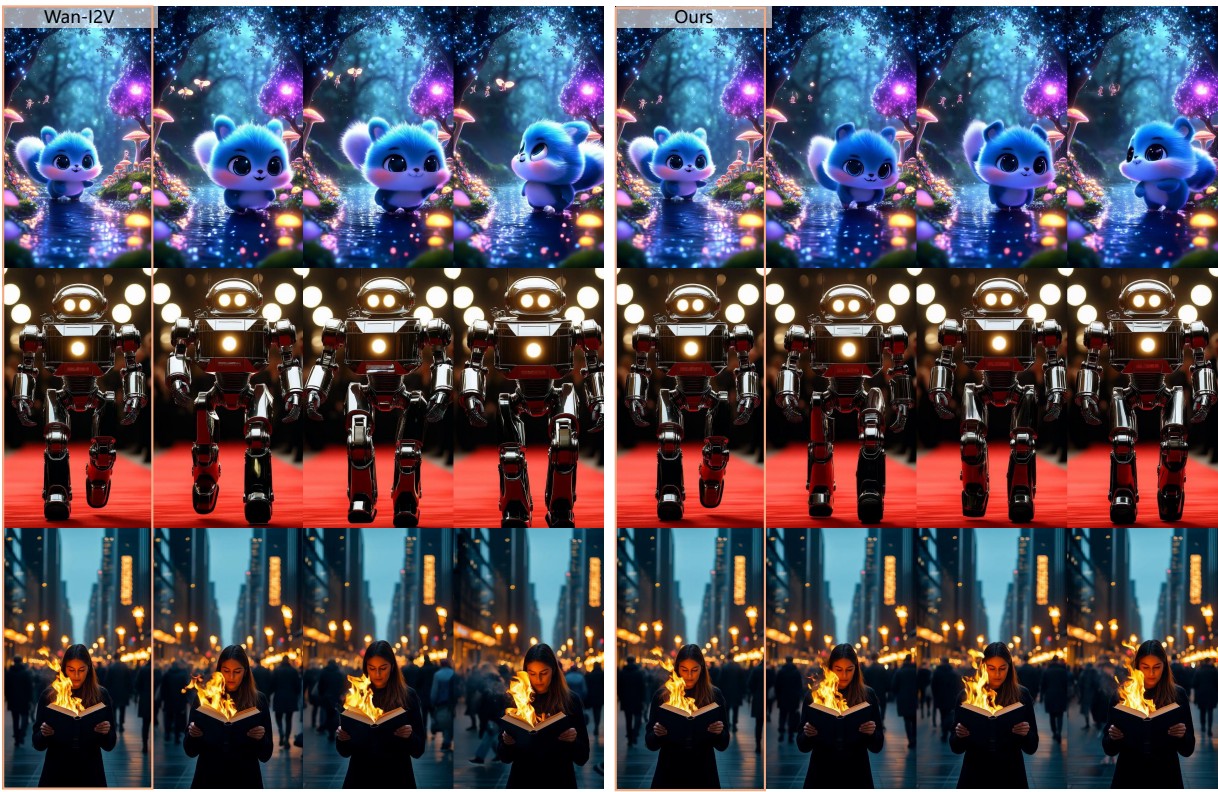

Figure 12: Distillation based on Wan-I2V. The first image in each example represents the conditioning image. Our method adapts well to the I2V task while maintaining comparable performance to the teacher model.

these risks, we advocate for the integration of digital watermarking into generated videos and support the development of deepfake detection techniques. We believe that the advancement of video synthesis research must go hand-in-hand with the progress of AI detection technologies to ensure responsible use.

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

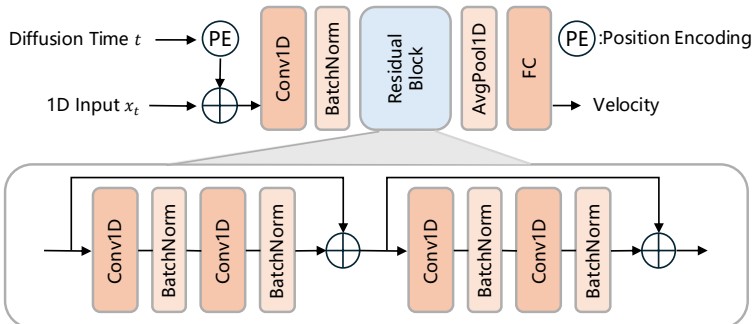

Figure 13: Model architecture for the 1D toy experiment.

Jintao Zhang, Jia Wei, Pengle Zhang, Jun Zhu, and Jianfei Chen. Sageattention: Accurate 8-bit attention for plug-and-play inference acceleration. In *ICLR*, 2025a.

Peiyuan Zhang, Yongqi Chen, Runlong Su, Hangliang Ding, Ion Stoica, Zhengzhong Liu, and Hao Zhang. Fast video generation with sliding tile attention. *arXiv preprint arXiv:2502.04507*, 2025b.

Xuanlei Zhao, Xiaolong Jin, Kai Wang, and Yang You. Real-time video generation with pyramid attention broadcast. In *ICLR*, 2025.

Zangwei Zheng, Xiangyu Peng, Tianji Yang, Chenhui Shen, Shenggui Li, Hongxin Liu, Yukun Zhou, Tianyi Li, and Yang You. Open-sora: Democratizing efficient video production for all. *arXiv preprint arXiv:2412.20404*, 2024.

Daquan Zhou, Weimin Wang, Hanshu Yan, Weiwei Lv, Yizhe Zhu, and Jiashi Feng. Magicvideo: Efficient video generation with latent diffusion models. *arXiv preprint arXiv:2211.11018*, 2022.

# A Appendix

## A.1 Details about the 1D Toy Experiment

In Sec. 3.2, we conduct a 1D toy experiment to analyze existing diffusion distillation methods. Specifically, we set the training data as $\{-3, 3\}$. The training loss is shown in Eq. (1). We use ResNet He et al. (2016) to construct our model, as illustrated in Fig. 13. The model is trained using the Adam optimizer for 10000 iterations with a learning rate of $10^{-4}$. Although our dataset consists of only two data points, we achieve a batch size of 2048 by repeating the data.

## A.2 SynVid

Table 5: Details resolutions about our synthetic dataset, SynVid.

| Resolution | $\mathbf{H} \times \mathbf{W} \times \mathbf{L}$ (Video) | $\mathbf{H} \times \mathbf{W} \times \mathbf{L}$ (Latents) | Number |
|---|---|---|---|
| High | 720×1280×129 | 90×160×33 | 8303 |
| Medium | 544×960×93 | 68×120×24 | 105996 |

Recent methods have demonstrated that fine-grained text prompts play a crucial role in video generation. Inspired by this, we leverage a multimodal large language model (MLLM), i.e., InternVL2.5-8B Chen et al. (2024c), to annotate real videos and obtain high-quality text prompts. As depicted in Fig. 14, the text prompts accurately describe the people and objects present in the videos, along with details about them, as well as dynamic motions. Moreover, they include descriptions of the atmosphere and background, which greatly aids in generating high-quality videos. The high-quality synthetic dataset facilitates the model distillation.

To promote community development and support future research, SynVid includes videos of different resolutions, *i.e.*, medium and high resolution, with detailed information provided in Tab. 5.

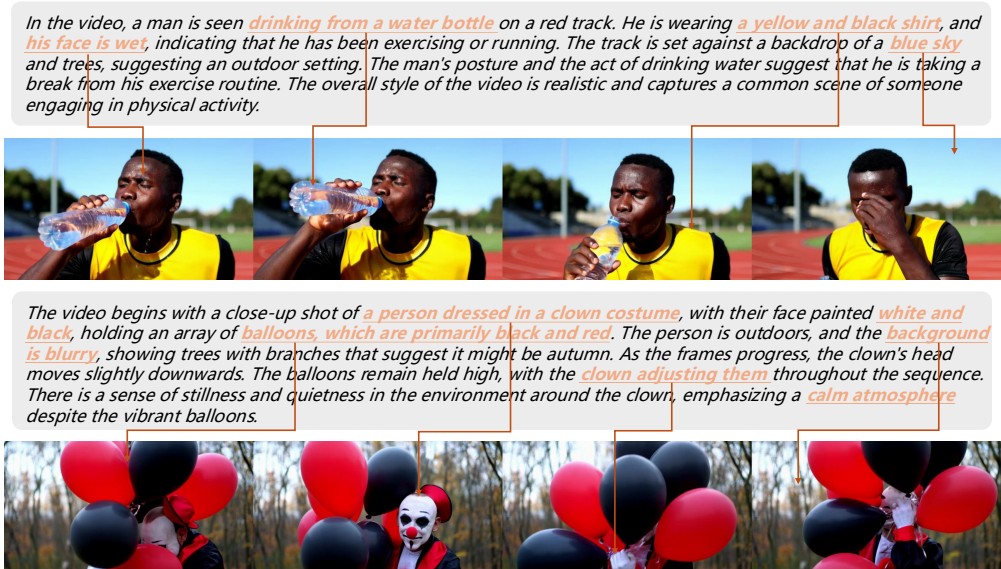

Figure 14: The illustration of our synthetic video dataset, SynVid.

## A.3 More Qualitative Results

Fig. 15 and Fig. 16 showcase the results of our method in generating videos at high and medium resolutions, respectively.

## A.4 Limitations and Future Work

Our method focuses on accelerating video diffusion models by distillation technique, which reduces the number of inference steps. However, generation speed is also influenced by VAE, which are used to encode and decode videos, making the acceleration of the VAE another promising direction for exploration HaCohen et al. (2024); Chen et al. (2025). Additionally, the DiT architecture contains many transformer blocks, and accelerating these transformer blocks is another area that warrants further investigation Wang et al. (2024b); Xie et al. (2025). We also analyze the failure modes of our distilled model. The primary artifact we observe is grid-like patterns in high-frequency details, such as water surfaces and foliage, where fine-grained structures are difficult to generate accurately with fewer inference steps. Mitigating high-frequency artifacts remains another important direction for future work.

*In a realistic close-up shot with smooth camera movement, **a charming woman is seen outdoors on a grassy lawn**. She is wearing a **white shirt** paired with a **white jacket**, and she adorns a necklace and earrings, adding elegance to her appearance. The woman is gracefully walking around an area enclosed by **a wooden fence**, moving in a gentle arc as she walks past the fence. The background features a lush green lawn and tent-like structures, creating a serene and refreshing atmosphere. The lighting is ample, highlighting the natural beauty of the scene.*

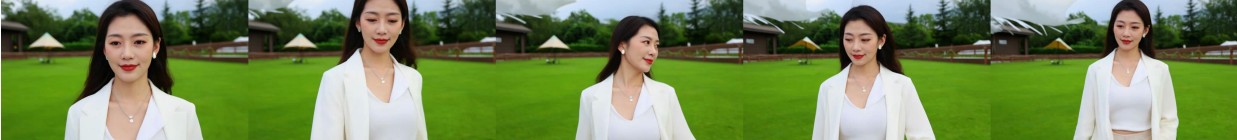

*A middle-aged sad **bald man** becomes happy as a wig of **curly hair and sunglasses** fall suddenly on his head.*

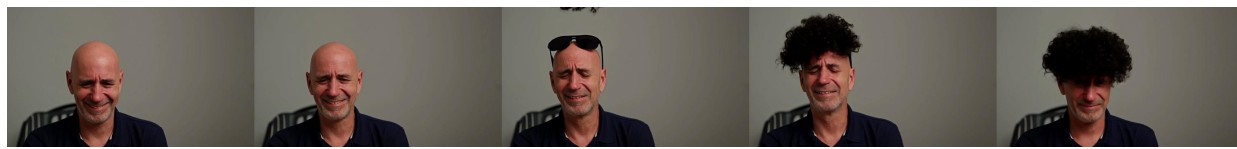

*A wide-angle view of a **dramatic cliffside** overlooking the ocean, **waves crashing** against the rocks far below.*

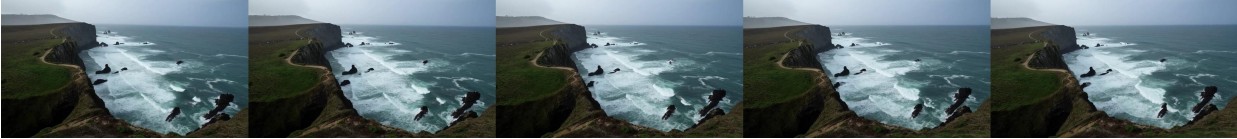

*A **dog wearing virtual reality goggles** in **sunset.***

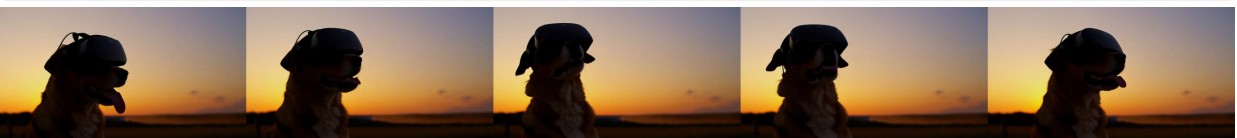

*A close-up of a **butterfly's wings**, showing the intricate patterns and vibrant colors in fine detail.*

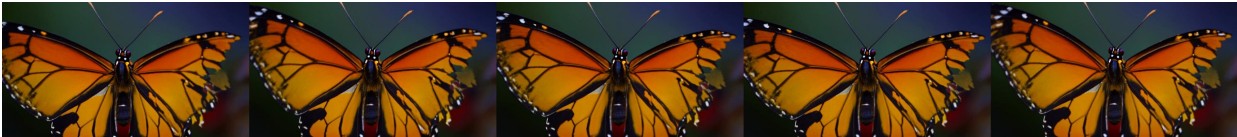

*A **capybara relaxes in a wooden barrel** filled with steaming hot spring water, its serene gaze adding tranquility to the scene. Perched atop its head is a **vibrant orange**, adding a playful contrast to its soft brown fur.*

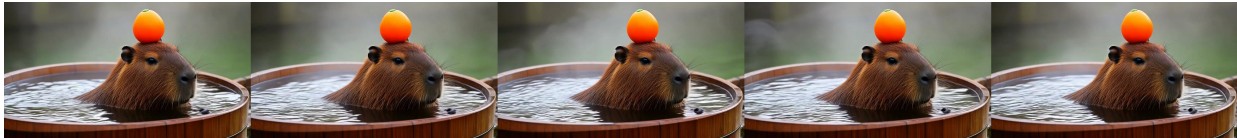

Figure 15: Qualitative results on text-to-video with high resolution, *i.e.*, 720×1280×129.

*A western **princess**, with **sunlight shining** through the leaves on her face, facial close-up.*

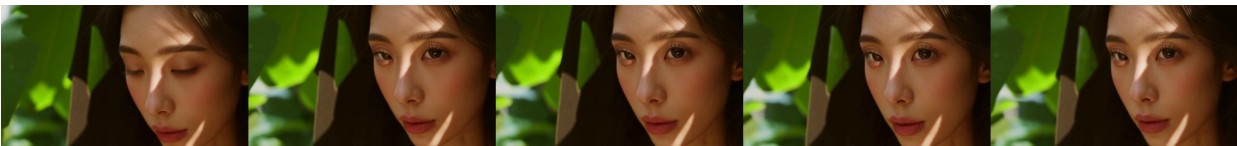

*A **beautiful woman** walking on the **school playground**. The sun shining on her face.*

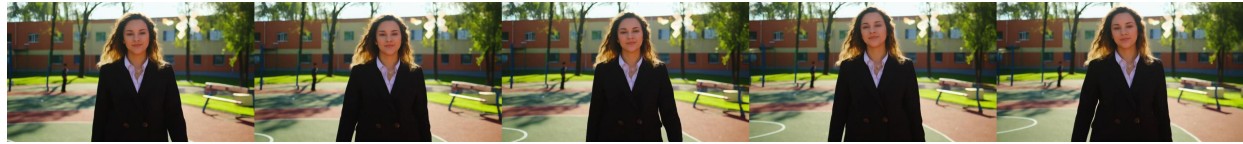

*An extreme close-up of an **gray-haired man** with **a beard** in his 60s, he is deep in thought pondering the history of the universe as he sits at a cafe in Paris, his eyes focus on people offscreen as they walk as he sits mostly motionless, he is dressed in a **wool coat suit coat** with a button-down shirt , he wears a brown beret and **glasses** and has a very professorial appearance, and the end he offers a subtle closed-mouth smile as if he found the answer to the mystery of life, the lighting is very cinematic with the golden light and the Parisian streets and city in the background, depth of field, cinematic 35mm film.*

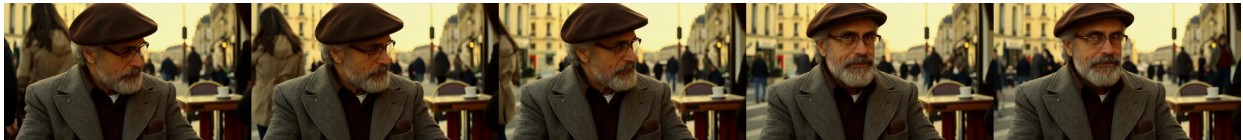

*A **white and orange tabby cat** is seen happily darting through **a dense garden**, as if chasing something. Its eyes are wide and happy as it jogs forward, scanning the branches, flowers, and leaves as it walks. **The path is narrow** as it makes its way between **all the plants**. the scene is captured from a **ground-level angle**, following the cat closely, giving a low and intimate perspective. The image is cinematic with warm tones and a grainy texture. The scattered daylight between the leaves and plants above creates a warm contrast, accentuating the cat's orange fur. The shot is clear and sharp, with a shallow depth of field.*

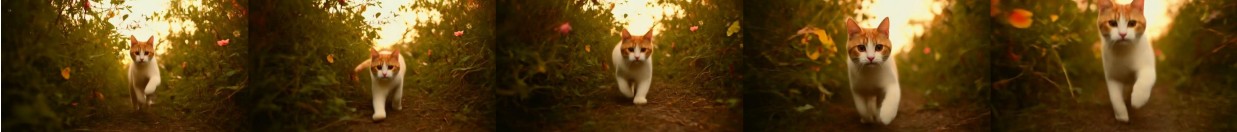

*The **monster stared at the food** with wide eyes and open mouth. Its posture and expression convey a sense of innocence and playfulness.*

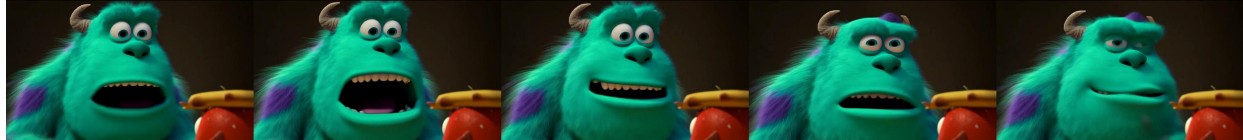

*A **panda in a scientist's lab coat**, conducting experiments with **beakers and test tubes**.*

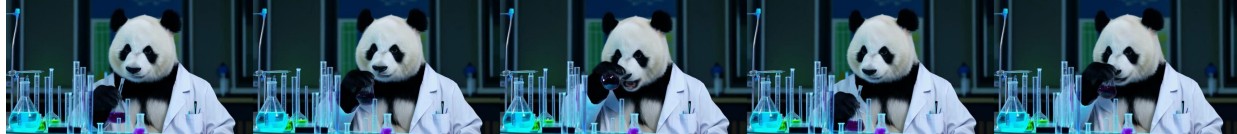

Figure 16: Qualitative results on text-to-video with medium resolution, *i.e.*, 544×960×93.

