# OpenReview forum: "AccVideo: Accelerating Video Diffusion Model with Synthetic Dataset"
_TMLR — Under review for TMLR_

### Review · Reviewer_9jVM · 2026-01-29

**Summary Of Contributions:**

The paper proposes a distillation-based approach to accelerate large video diffusion models, achieving approximately 6–8× faster inference compared to the teacher models. The method targets pretrained video diffusion models such as HunyuanVideo and Wan.
The authors begin with an analysis of diffusion distillation and argue that commonly used distillation procedures may rely on ineffective or off-trajectory (useless) data points, which can lead to distribution mismatch between teacher and student models.
Based on this observation, they propose a trajectory-based distillation scheme that restricts supervision to intermediate latent states sampled directly from the teacher’s denoising trajectories.
To support this training paradigm, the authors construct a synthetic dataset generated entirely from the teacher model, where both final outputs and intermediate denoising latents are stored.
Using this dataset, the student model is trained to map between sparsely selected timesteps along the denoising trajectory, enabling few-step generation.
In addition to the trajectory-based loss, the method incorporates an adversarial training objective to align the student’s intermediate latent distributions with those of the teacher at selected timesteps. The discriminator leverages frozen teacher features, following prior adversarial diffusion distillation works.
Experimental results on large-scale video generation tasks show that the distilled models significantly reduce inference cost while largely preserving visual quality and temporal coherence. The approach is demonstrated on multiple teacher models and supports both text-to-video and image-to-video generation.

**Strengths**
- The method is clearly described, with sufficient algorithmic and implementation details to support reproducibility.
- The empirical results demonstrate strong practical performance, achieving substantial speedups while maintaining competitive video quality.
- The approach is validated across multiple large pretrained video diffusion models, suggesting reasonable generality.

**Weaknesses**

1. The paper frames its approach as novel, but many of its components closely resemble existing diffusion distillation methods. In particular, trajectory-based distillation has been widely explored distillation approach in prior work, including progressive distillation (Salimans & Ho, 2022), consistency models (Song et al., 2023), and recent video-specific methods mentioned in the paper such as CausVid (Yin et al., 2024) and APT (Lin et al., 2025).
The primary distinction appears to be the explicit storage and reuse of intermediate denoising trajectories as a synthetic dataset, which is a relatively minor contribution.

2. The choice of sparse timesteps used for trajectory-based distillation is adopted from prior work and justified empirically, but the paper does not provide sufficient intuition or analysis to make this design choice self-contained.

**References**

Salimans, Tim, and Jonathan Ho. "Progressive distillation for fast sampling of diffusion models." ICLR (2022).

Song, Yang, et al. "Consistency models." ICML (2023).

**Additional Comments:**

- After equation "3" the paper mentions "through the first-order Euler Ordinary Differential Equation (ODE) solver.", but this is too restrictive. It should be possible to use other first-order ODE solvers, not just Euler.

- The arrangement of subplot is in Figure 2 is confusing.

- The caption in Table 2 "The results marked with ∗ are tested by us,". However none of the results has *.

**Audience:**

Yes

**Audience Explanation:**

One of the primary limitations of large video diffusion models is their high computational cost at inference time, and this work directly addresses that bottleneck.
The proposed distillation framework demonstrates that substantial inference speedups can be achieved for large-scale video generation models while maintaining competitive visual quality, which is likely to be of interest to researchers working on efficient generative modeling.

Although the conceptual novelty of the approach is limited, the paper provides a practical and stable recipe for combining trajectory-based distillation with adversarial training in the context of large video diffusion models. This integration may be valuable to future work on model acceleration and could serve as a useful reference or baseline for subsequent distillation-based approaches.

**Broader Impact Concerns:**

In video generation, there are always ethical concerns about generating fake misleading material, or deepfakes, which is not addressed in the paper.
I suggest adding a Broader Impact Statement to the paper to address these ethical implications.

**Claims And Evidence:**

Yes

**Claims Explanation:**

Overall, the paper provides sufficient empirical evidence to support its main claims.
The proposed approach is evaluated on two large and widely used video diffusion models, HunyuanVideo and Wan, which supports the generalizability across different architectures and training setups.
The experimental results demonstrate consistent inference speedups while largely preserving generation quality, and the ablation studies help justify the contribution of individual components of the method.

That said, while the empirical evidence is generally convincing, some design choices, such as the selection of distillation timesteps, are justified primarily by prior work rather than by direct analysis in this paper. Providing additional evidence or intuition for these choices would further strengthen the support for the claims.

**Requested Changes:**

1. Clearly state what is actually novel in the proposed approach. If the contribution mainly lies in combining existing techniques, this should be stated explicitly. The paper should better position itself within prior work on accelerating video diffusion models, for instance, in "Accelerating Video Diffusion Models." in Section 2.

2. The choice of distillation timesteps is not sufficiently justified and currently relies on another paper. A brief explanation or ablation should be added to make the method self-contained.

---

> ### Author Response · Authors · 2026-04-30
> **Rebuttal by authors**
>
> We sincerely thank the reviewer's thorough and constructive feedback. We address your constructive comments below:
>
> > **W1:** The paper frames its approach as novel, but many of its components closely resemble existing diffusion distillation methods...... & **R1:** Clearly state what is actually novel in the proposed approach......
>
> **A1:** We acknowledge that the general idea of trajectory-based distillation has been explored in prior methods. However, we would like to clarify the key distinctions of our method. Existing methods that employ trajectory-based distillation typically rely on real data combined with forward diffusion to obtain the start points $x_t$ of each distillation step. As analyzed in Sec.3.2, this inevitably introduces two types of useless data points, which causes the teacher model to provide unreliable guidance to the student, adversely affecting the performance of the student (as shown in Table.3 in the revised version). Our contribution is providing a detailed analysis of these issues and leveraging the synthetic dataset to eliminate the useless data points during distillation. Beyond the synthetic dataset, our work further explores the rich prior encoded in intermediate denoising trajectories. Specifically, we propose adversarial training that explores the distribution information captured at each diffusion timestep, and investigate the behavior of hidden states at different timesteps, which further improves the distillation performance as shown in Fig.8. We have revised our paper to more explicitly articulate these distinctions from existing methods in Sec.2.
>
> > **W2:** The choice of sparse timesteps used for trajectory-based distillation...... & **R2:** The choice of distillation timesteps is not sufficiently justified and currently relies on another paper......
>
> **A2:** Thanks for your constructive suggestion. To provide direct empirical evidence, we conducted an additional ablation using a uniform timestep grid selected from our synthetic trajectories: $t = \{1000.0, 810.9, 605.7, 378.3, 225.8, 0.0\}$. The results on VBench are reported below:
>
> | **Timestep Schedule** | **Total Score** | **Quality Score** | **Semantic Score** |
>   |---|---|---|---|
>   | Uniform grid-544P | 82.86% | 84.33% | 76.98% |
>   | Ours-544P | **83.26%** | **84.58%** | **77.96%** |
>
> The results demonstrate that our timestep schedule outperforms the uniform grid, validating the design choice empirically. We hypothesize that the non-uniform schedule, which allocates more timesteps in the high-noise region, better accommodates the varying learning difficulty across diffusion timesteps. We include this ablation in the revised version.
>
> > **Broader Impact Concerns**
>
> **A3:** We thank the reviewer for raising this concern. We have added a Broader Impact Statement section in the revised version. We acknowledge that our method significantly reducing the computational cost of high-quality video generation, it could potentially be misused for generating deepfakes or other misleading content. To mitigate these risks, we advocate for the integration of digital watermarking and support the development of deepfake detection techniques. We believe that the advancement of synthesis research must go hand-in-hand with the progress of AI detection technologies to ensure responsible use.
>
> > **C1:** After equation "3" the paper mentions "through the first-order Euler Ordinary Differential Equation (ODE) solver."......
>
> **A4:** Thank you for these careful observations. Our method is built upon the Flow Matching framework and is compatible with other first-order ODE solvers. We have clarified this in the revised paper.
>
> > **C2:** The arrangement of subplot is in Figure 2 is confusing.
>
> **A5:** We apologize for the confusion and would like to explain the intended logic of the layout. The figure is organized as follows. Top-Left illustrates the setup of our 1D toy experiment. Top-right (a,b,c,d) illustrates the pipelines and problems of existing distillation methods. Middle (e) shows that useless data points exist in practice, by measuring their frequency using the mismatching degree $M$. Bottom row (f) demonstrates that useless data points during distillation degrade the performance, motivating our method. We have added an improved caption in the revised version.
>
> > **C3:** The caption in Table 2 "The results marked with ∗ are tested by us,". However none of the results has *.
>
> **A6:** We apologize for the oversight. $*$ in Table 2 refers to the results of HunyuanVideo-544P, Ours-544P, and Ours-720P, which were evaluated by us. We have fixed the mistake in the revised version.

---

### Review · Reviewer_y7b3 · 2026-02-11

**Summary Of Contributions:**

AccVideo addresses the inefficiency of video diffusion models by identifying that previous distillation methods rely on "useless data points" that deviate from the teacher's actual denoising trajectories. To solve this, the authors propose a novel distillation framework using SynVid, a synthetic dataset of 110K high-quality videos and their corresponding valid denoising trajectories. They implement a trajectory-based few-step guidance that enables the student model to learn a direct noise-to-video mapping, alongside an adversarial training strategy with timestep-aware projection heads to align intermediate data distributions and enhance video quality.

**Audience:**

Yes

**Audience Explanation:**

While the core methodology of this paper is largely recognized within the diffusion model community, e.g., the emphasis on "training-sampling mismatch" already highlighted by DMD2, and the use of "teacher trajectories" established by Rectified Flow, its practical implementation for the specific pain point of video acceleration still offers value to a subset of the audience as some Video-Specific Feature Tuning, it analysis of different layers within the video model (e.g., layers 40 and 60) for capturing fine-grained details, combined with the design of timestep-aware projection heads, provides a specific engineering guide for handling the unique feature distributions of video data.

**Broader Impact Concerns:**

n.a.

**Claims And Evidence:**

Yes

**Claims Explanation:**

The claims in the submission are supported by accurate and convincing evidence through a combination of quantitative benchmarks, qualitative visual results, and targeted ablation studies.

The authors provide empirical proof of an 8.5x improvement in generation speed (reducing high-resolution inference from 3234s to 380s)。 Furthermore, the technical necessity of their "synthetic trajectory" approach is validated by a 1D toy experiment and detailed ablation studies, which confirm that both trajectory guidance and adversarial training are essential for eliminating artifacts and stabilizing the distillation process.

**Requested Changes:**

Many indicators and baseline are currently outdated. I suggest that the author update the baseline in the ablation section to match the latest progress.

---

> ### Author Response · Authors · 2026-04-30
> **Rebuttal by authors**
>
> Thank you for recognizing and valuing our work. We address your constructive comments as follows:
>
> > **R1:** Many indicators and baseline are currently outdated. I suggest that the author update the baseline in the ablation section to match the latest progress.
>
> **A1:** We thank the reviewer for this suggestion. To provide a stronger ablation baseline, we conducted an additional experiment using the FastVideo framework with real data and DMD as the distillation objective, built upon the same HunyuanVideo teacher model. The inference step is kept identical to ours (5 steps) to ensure a fair comparison. The VBench results are reported below:
>
> | **Method** | **Total Score** | **Quality Score** | **Semantic Score** |
>   |---|---|---|---|
>   | Real data + DMD | 82.95% | 84.23% | 77.86% |
>   | Ours | **83.26%** | **84.58%** | **77.96%** |
>
> Training with real data and DMD leads to inferior performance compared to our method, demonstrating the importance of leveraging synthetic data and our proposed distillation method. We have included this ablation in the revised version.

---

### Review · Reviewer_z9js · 2026-04-20

**Summary Of Contributions:**

The paper proposes **AccVideo**, a diffusion-distillation method that accelerates large pretrained video diffusion models (HunyuanVideo-13B, Wan-14B) by training a student to traverse a small number of key points along *teacher-generated denoising trajectories*. The central contributions are:

1. **Conceptual framing**: a 1D toy analysis attributing the quality loss of prior distillation methods to "useless data points", i.e. points that lie off the teacher's denoising trajectory due to dataset mismatch or Gaussian-noise mismatch during forward diffusion.
2. **SynVid dataset**: 110K (synthetic video, full 50-step denoising trajectory, fine-grained caption) tuples generated by HunyuanVideo with InternVL2.5-8B captions; two resolution tiers (544×960×93 and 720×1280×129).
3. **Trajectory-based few-step guidance**: trains the student to predict the finite-difference velocity between adjacent *key timesteps* {t'_5=1.0, 0.9651, 0.9130, 0.8235, 0.6363, t'_0=0} selected on each teacher trajectory.
4. **Adversarial training with timestep-aware heads**: a GAN loss applied at each key timestep using the frozen teacher as a feature extractor (layer 60 for noisy steps, layer 40 for t=0) and separate projection heads per timestep; a replay queue avoids the prohibitive cost of re-sampling student trajectories.
5. **Empirical validation**: 8.5× speedup over HunyuanVideo-720P (3234 s → 380 s on A100) with VBench total-score changes of −0.47% at 720P and +0.59% at 544P; extension to Wan-T2V-14B (9.6×) and Wan-I2V-14B (6.8×).

**Audience:**

Yes

**Audience Explanation:**

The video diffusion community would be interested.

**Broader Impact Concerns:**

No conerns

**Claims And Evidence:**

No

**Claims Explanation:**

Some are well supported for example: The speed claim is backed by clear wall-clock numbers on a consistent A100 setup (Table 1). The transferability claim  is also clearly evidenced by the paired evaluations in Sec. 5.3.

Some are not. For example, the authors claim the model maintains comparable performance as baseline methods but results in Table 2 show a non-neglectable drop in performance. Ablation is not sufficient enough. For example, the choice of the times seem artificial and, while justified, but not compared with other obvious alternatives like uniform grid, etc.

**Requested Changes:**

1. The "useless data points" are discussed in the toy example but I am curious how does they translate to real applications like video generation? There is no measurement of the frequency or effect of useless data points in the actual video-distillation setting.
2. What is the cost to generate the dataset? Although training can be done in 12 days with 8 A100 GPUs, generating such dataset offline still seem prohibitively costly.
3. Ablation seems sparse. For example, an ablation on the choice of the time steps may be helpful.
4. Would a analysis on failure modes be possible? There is no discussion of prompts where AccVideo underperforms the teacher, or where subject-consistency loss becomes visible (which it should, given the 3.17% drop at 720P).

---

> ### Author Response · Authors · 2026-04-30
> **Rebuttal by authors (Part 1)**
>
> Thank you for recognizing and valuing our work. We address your constructive comments as follows:
>
> > **Claims1:** The authors claim the model maintains comparable performance as baseline methods but results in Table 2 show a non-neglectable drop in performance. & **R4:** Would a analysis on failure modes be possible......
>
> **A1:** We appreciate this comment and would like to clarify the source of the performance gap, and provide an analysis of the failure modes. It is important to note that our student model is distilled at 544P resolution as described in Implementation Details. As shown in Tab.2, at this resolution our method achieves a Total Score of 83.26\%, superior to the teacher model's 82.67\%. More specifically, Subject Consistency is comparable (94.46\% vs. 94.33\%), demonstrating that our distilled model preserves the generation quality of the teacher at the distillation resolution. The gap at 720P (from 83.24\% to 82.77\%) stems from the resolution mismatching between the distillation resolution (544P) and the evaluation resolution (720P). When the distilled model is applied to a higher resolution than it was distilled on, generation quality is sacrificed. To further verify this, we conduct an additional experiment where we distill the model directly at 720P. The results on VBench are reported below (for complete results, please refer to our revised version):
>
> | **Method** | **Total Score** | **Quality Score** | **Semantic Score** | **Subject Consistency** |
>   |---|---|---|---|---|
>   | HunyuanVideo-720P (teacher, 50 steps) | 83.24% | 85.09% | 75.82% | 97.37% |
>   | Ours-720P (distilled at 544P, 5 steps) | 82.77% | 84.38% | 76.34% | 94.20% |
>   | Ours-720P (distilled at 720P, 5 steps) | 83.21% | 84.94% | 76.30% | 97.32% |
>
> When distilled directly at 720P, our method achieves a Total Score of 83.21\%, which is comparable to the teacher model (83.24\%), confirming the gap is attribute to the resolution mismatching. Notably, Subject Consistency improves from 94.20\% (distilled at 544P) to 97.32\% (distilled at 720P), closing the gap with the teacher model (97.37\%).
>
> We further analyzed the failure modes of our distilled model. The primary artifact we observe is grid-like patterns in high-frequency details, such as water surfaces and foliage, where fine-grained structures are challenging to generate accurately with fewer inference steps. We have added a dedicated failure mode analysis to the Limitation in the revised version.
>
> > **Claims2:** Ablation is not sufficient enough & **R3:** Ablation seems sparse. For example, an ablation on the choice of the time steps may be helpful.
>
> **A2:** We appreciate this suggestion and would like to provide additional ablation. To provide direct empirical evidence, we conducted an additional ablation using a uniform timestep schedule selected from our synthetic trajectories: $t = \{1000.0, 810.9, 605.7, 378.3, 225.8, 0.0\}$. The results on VBench are reported below:
>
> | **Timestep Schedule** | **Total Score** | **Quality Score** | **Semantic Score** |
>   |---|---|---|---|
>   | Uniform grid-544P | 82.86% | 84.33% | 76.98% |
>   | Ours-544P | **83.26%** | **84.58%** | **77.96%** |
>
> The results demonstrate that our timestep schedule outperforms the uniform grid, validating the design choice empirically. We hypothesize that the non-uniform schedule, which allocates more timesteps in the high-noise region, better accommodates the varying learning difficulty across diffusion timesteps. We have included this ablation in the revised version.

---

> ### Author Response · Authors · 2026-04-30
> **Rebuttal by Author (Part 2)**
>
> > **R1:** The "useless data points" are discussed in the toy example......
>
> **A3:** We appreciate this comment and would like to clarify why direct frequency measurement in the actual video-distillation setting is non-trivial, and then provide empirical evidence for the harmful effect of useless data points. In the actual video-distillation setting, determining whether a data point lies on a valid denoising trajectory requires access to the complete trajectory distribution of the teacher model. However, it is difficult to fully generate all trajectories. Therefore, it is not feasible to directly measure the frequency of useless data points.
>
> While direct frequency counting is infeasible, we provide two experiments that demonstrate the negative impact of useless data points. In Tab.3, our ablation comparing **Real data + PCM** (renamed from ``Distill on real'' for clarity) shows that training with synthetic data yields a clear improvement in Total Score (79.52\% -> 83.26\%), demonstrating the benefit of eliminating useless data points through our synthetic dataset. Moreover, we conducte an additional ablation using the FastVideo framework with real data and DMD distillation under same settings (HunyuanVideo teacher, 5 inference steps):
>
> | **Method** | **Total Score** | **Quality Score** | **Semantic Score** |
>   |---|---|---|---|
>   | Real data + DMD | 82.95% | 84.23% | 77.86% |
>   | Ours | **83.26%** | **84.58%** | **77.96%** |
>
> It confirms that training with real data leads to inferior performance compared to our method, providing evidence that useless data points harm distillation quality in practice.
>
> > **R2:** What is the cost to generate the dataset......
>
> **A4:** We apologize for the omission and provide the details. Specifically, generating the full SynVid dataset takes approximately 7K GPU-hours on A100 GPUs. Here, we leverage sequence parallel to accelerate the generation time.  Importantly, although generating the complete dataset is time-consuming, our experiments indicate that strong distillation performance can already be achieved with a relatively small subset of synthetic data (38.4K samples), which substantially reduces the practical cost of data generation.

---

### Author Response · Authors · 2026-04-30
**Summary of rebuttal**

We sincerely thank all three reviewers for their time and constructive feedback. We are grateful that the reviewers recognized our work. In particular, Reviewer z9js acknowledged our claims regarding "speed and transferability"; Reviewer y7b3 found our claims to be supported by "accurate and convincing evidence"; and Reviewer 9jVM noted that our paper provides "sufficient empirical support for its main claims". Moreover, all reviewers agreed that our work would be of interest to the TMLR audience.

In response to the reviewers' concerns, we have made the following updates:

(We have addressed all reviewer concerns and incorporated the suggested additional experiments in our revised manuscript. All revised content is highlighted in $\color{blue}{\text{blue}}$.)

- **Performance drop and failure modes** (z9js-Claims1/R4): We clarified that the performance drop at 720P stems from resolution mismatch, and added the primary failure mode to the Limitation section.

- **Timestep schedule** (z9js-Claims1/R3, 9jVM-W2/R2): We added a new ablation experiment comparing against a uniform timestep grid, with results reported on VBench.

- **Updated ablation baseline** (y7b3-R1): We added a new ablation experiment using the FastVideo framework with real data and DMD distillation under identical settings, demonstrating the superiority of our method.

- **Useless data points** (z9js-R1): We explained why direct frequency measurement is infeasible in practice, and provided two experiments demonstrating the harmful effect of useless data points on distillation performance.

- **Broader Impact Statement** (9jVM): We have added a Broader Impact Statement section discussing the risks and mitigation strategies in the revised paper.

- **Other Issues** (9jVM-C1/C2/C3, z9js-R2): We have refined the paper to improve the clarity of both the text and figures, and have disclosed the dataset generation cost.